# A Study on mPPO Development and Injection Molding Process for Lightweight Stack Enclosure of FCEV

**DOI:** 10.3390/polym15051303

**Published:** 2023-03-04

**Authors:** Soo-Lim Lee, Jong-Hwal Kim, Seon-Bong Lee

**Affiliations:** 1Department of Mechanical Engineering, Keimyung University, Daegu 42601, Republic of Korea; 2Desco, Chilgok-Gun, Kyungbuk 39909, Republic of Korea; 3Division of Mechanical and Automotive Engineering, Keimyung University, Daegu 42601, Republic of Korea

**Keywords:** FCEV stack enclosure, mPPO, injection molding analysis, Taguchi method

## Abstract

The automobile industry is focused on eco-friendly vehicles with the goal of carbon neutrality (Netzero), and vehicle weight reduction is essential to achieve high fuel efficiency for driving performance and distance compared to internal combustion engines. This is important for the light-weight stack enclosure of FCEV. Moreover, mPPO needs to be developed with injection molding for the replacement of existing material (aluminum). For this purpose, this study develops mPPO and presents it through physical property tests, predicts the injection molding process flow system for stack enclosure production, proposes injection molding process conditions to secure productivity, and verifies conditions through mechanical stiffness analysis. As a result of the analysis, the runner system with pin-point gate and tab gate’s sizes are proposed. In addition, injection molding process conditions were proposed with the results of cycle time 107.627 s and reduced weld lines. As a result of the strength analysis, it can withstand the load 5933 kg. Therefore, it is possible to reduce weight and material costs using the mPPO existing manufacturing process with existing aluminum, and it is expected that there would be effects, such as reducing the production cost by securing productivity through reducing cycle time.

## 1. Introduction

The commitment to carbon neutrality continues worldwide under the recently applicable New Climate Regime, requiring low-carbon strategies for energy production and use with the United Nations Framework Convention on Climate Change (UNFCCC) demanding a transition to an economic structure that reduces greenhouse gases. In the field of eco-friendly vehicles, which is one of the major areas to resolve these social issues, there is increasing attention to the technology development for eco-friendly vehicles, such as hydrogen fuel-cell electric vehicles (FCEV) [1]. As a key means for realizing carbon neutrality in the automobile business, the hydrogen fuel cell (HFC) is being developed as a core powertrain similar to a locomotive engine. However, current fuel cell stack enclosures using aluminum alloys are manufactured by a die-casting method, which is experiencing difficulties in mass production due to reduced productivity.

In addition, massive efforts are directed on the development of FCEV technologies, mainly aiming at high fuel economy and efficiency, and vehicle light-weighting is drawing attention to enhance fuel mileage and driving performance. In the case of a FCEV, the reduction in the weight of a passenger car by 10% enhances fuel economy by 4~6%, resulting in improved acceleration performance, shortened braking distances, and increased durability of components, such as the chassis. Therefore, lightweight automotive parts, such as hydrogen fuel cell stack enclosures, are under active development, and for this purpose, there is an increasing attention to the development of engineering plastics with high-strength mechanical properties as a new material that can replace the stability of existing metal materials. Engineering plastics emerged in the 1930s when Company D in the United States produced PA66 (Polyamide, Nylon), a material for textiles. A subsequent release of PC (Polycarbonate) and POM (Polyoxymethylene, Polyacetal) formed the market, and PBT (Polybutylene Terephthalate) and mPPO (modified Polyphenylene Oxide, modified Polyphenylene Ether) developed in the 1970s, became dominant in the market as one of the five major engineering plastics. The corresponding material is applied to bumpers, and an application of injection molding can secure productivity with a shorter cycle time compared to the production of existing materials.

Regarding the development of mPPO, Won et al. studied the physical properties of impact strength and thermal expansion behavior according to fiber orientation by reinforcing short carbon-fibers in PPO/PA6 composite material [2], and Chandra et al. prepared PPO/PET composites, observed the molecular structure, and studied physical properties such as mechanical strength [3]. Lee et al. verified the potential as a composite material by measuring the electrical and mechanical properties of a PA66/PPO composite by adjusting the amount of multi-walled carbon tubes (MWCNT) [4]. Lee et al. investigated the phase structure and rheological properties of a PA/PPO composite using graphene oxide (GO) according to mixing conditions [5], and Habaue et al. synthesized PPO with oxidative coupling polymerization using a CuCl (Copper chloride)/Amine catalyst system and verified the improvement in physical properties caused by the polymerization method [6]. In addition, Ahn et al. studied the manufacturing of composites by adding glass-fibers (GF) to mPPO alloyed with PS and PA [7], and Liu et al. proposed mPPO composed of PPO and high-impact poly styrene (HIPS) whose dielectric loss resistance, heat resistance, and mechanical properties are improved by adding MgTiO_3_-Ca0.7La0.2TiO_3_ (MTCLT) ceramic [8]. Li et al. proposed resin, alloyed with epoxy and PPO, that meets the requirements of prepreg, which is an intermediate stage of composite materials, in terms of thermal stability, fluidity, mechanical properties, and shape [9]. Xie et al. improved the stiffness and interfacial adhesion by surface-reforming a PPO/PS composite using in situ copolymerization, and studied the tendency in relation to the polymerization method [10].

Regarding the study proposing the injection molding process using Computer Aided Engineering (CAE), Saedon et al. constructed a 3D CAD model and performed an injection molding simulation on a disposable oral mirror product using polypropylene (PP), and suggested mold temperature, melting temperature, and cooling time to minimize defects [11]. A pressure-based system was developed to measure the pressure change over time and to see whether it met the optimal conditions, and an experimental approach was considered for the case of constant filling speed and packing pressure. Ozcelik et al. modeled a thin surface cover made of PC and acrylonitrile butadiene styrene (ABS), selected injection parameters for warpage, verified through analysis with Moldflow software, and proposed that the packing pressure was the greatest influence on warpage [12]. Huang et al. investigated factors that have a high influence on warpage in thin-walled molded products produced by injection molding, using C-Mold software [13]. Shenet et al. analyzed parts used in the electronic and computer industries by applying mold temperature, injection temperature, injection time, gate location, etc. based on MPI software [14].

Regarding the study of selecting injection process conditions using the design of experimental method (DOE), Stanek et al. proposed ideal conditions using the DOE method to optimize the injection molding process, as well as observed, and compared the dimensions, shapes, and characteristics of molded products [15], and Mohan et al. analyzed the post-processing shrinkage and warpage of parts molded with PP and investigated the effects of injection molding process variables using the DOE [16]. Jou et al. proposed process conditions for minimizing shrinkage of injection-molded products including optical-fibers using the Taguchi DOE and surface-reaction method [17], and Mehdi et al. selected resin melt temperature, gate system design, filling and cooling time as major process parameters according to the Taguchi DOE and fuzzy analysis hierarchical method, evaluated general manufacturing defects, such as short-shots, and found the optimal method to meet the mold-ability [18].

To replace the existing metal material, it must have excellent mechanical strength and dimensional stability. In the case of mPPO, it has excellent heat and chemical resistance, as well as excellent low water absorption, and is unrivaled in dimensional stability. Although it has the disadvantage of poor formability, the part can be improved through engineering plastics with good formability, and its mechanical properties, thermal properties, and electrical properties are excellent. Due to these advantages, mPPO is currently being produced in the automobile industry by applying mPPO. However, the physical properties of mPPO vary depending on the type of mixed material since different types of engineering plastics are mixed with PPO, and there has been insufficient research in developing mPPO to replace the existing metal materials of the stack enclosure of FCEV. Therefore, the aim of this study is to investigate injection molding process conditions for enclosures with required mechanical stiffness by developing mPPO with applicable characteristics for lightweight stack enclosures while minimizing production cycle time. For this purpose, this study will develop mPPO and present it through physical property tests, predict the injection molding process flow system for stack enclosure production, propose injection molding process conditions to secure productivity, and verify conditions through mechanical stiffness analysis, as presented in Figure 1.

## 2. Theoretical Background

### 2.1. Development of mPPO

Plastic refers to a polymeric compound that can be molded by deformation with heat or pressure, and general-purpose plastics tend to break under external force in contrast to metal materials. The plastics with enhanced mechanical stiffness and heat resistance are classified as high-performance plastics, and materials with a glass transition temperature of 100 °C or higher are called engineering plastics [19].

Engineering plastics typically include polycarbonate, polyamide, and mPPO. Especially in the case of mPPO, it is also used to manufacture automobile parts and specializes in low moisture absorption [20]; therefore, it has excellent electrical insulation compared to other engineering plastics and unrivaled dimensional stability [21]. For mechanical properties, molding shrinkage is minor, dimensional accuracy is excellent, high specific heat results in a slight change in its mechanical properties due to temperature change, creep at a meager rate, and superior tensile strength. For thermal properties, its thermal deformation temperature is under high loads ranging from 90 to 170 °C, which gives it heat-resisting applications, and its coefficient of linear thermal expansion (CLTE) is close to the level of metal materials among engineering plastics, making it suitable as a structural material for precision functional parts. For electrical properties, its permittivity and dissipation factors are tiny, it remains strong under different frequencies, temperatures, and humidity levels, and its volume resistivity and dielectric strength are excellent. Additionally, PPOs are not affected by most acids, alkalis, or organic solvents. Despite these excellent properties, PPOs have disadvantages in that they are not suitable for processing due to their high molding temperatures and poor fluidity when being molded into products [22]. Since mPPO is prepared by mixing PPO resin with another resin, it is called modified PPO, and it has the characteristic that, depending on the mixing ratio, physical properties suitable for automobile parts can be developed. PPO is a thermoplastic resin with excellent mechanical stiffness, dimensional stability, heat resistance, and electrical insulation, which makes it suitable for bumpers and battery cases [23]. However, since it is difficult to be independently applicable to processing products due to poor mold-ability in contrast to excellent physical properties, a previous study investigated the mixing ratio of mPPO suitable for injection molding by mixing PPO (80%, 70%, 60%, 50%, 40%, 30%, 20%) with PA66 (20%, 30%, 40%, 50%, 60%, 70%, 80%), which has excellent mold-ability, thereby proposing 40% of PPO and 60% of PA66 as an optimal ratio, with which no unfilled enclosure shape occurs [20].

However, to upgrade the mPPO (PPO 40%/PA66 60%) proposed in the previous study, which has relatively excellent mechanical stiffness but falls behind metals, this study conducts a comparative evaluation of a new material, to which 30% of glass fiber, a type of filler, is added to enhance mechanical stiffness. Glass fiber was selected since the advantage of improving both strength and toughness is generally known and is relatively inexpensive and easy to obtain [24].

Glass fiber-reinforcement reduces shrinkage and improves stiffness in plastics by limiting the movement of polymer chains, as well as improving impact resistance by reducing stress cracking. Furthermore, the heat resistance is excellent due to the increase in temperature, and the materials became fire resistant due to their reduced combustibility.

However, reinforcement can result in reduced tensile strength and increased brittleness, and fluidity is lowered due to an increase in melt viscosity, which can be disadvantageous during injection molding.

Therefore, there is a need for comparing the physical properties of non-reinforced mPPO mixed with 40% PPO and 60% PA66 and reinforced mPPO with the addition of glass fibers.

### 2.2. Injection Molding Process and Its Major Factors

Injection molding is a molding process for manufacturing plastics, and the process conditions vary depending on the shape and material characteristics. Injection molding refers to the process of plasticizing resins, filling, and packing the molten resin into the mold cavity of the product shape using pressure, and then ejecting the molded product after a certain period of cooling. One cycle is defined from the plasticizing stage to the ejection stage [25].

Major injection molding process variables for each step include resin melting temperature, mold temperature, filling time, switchover point, packing, cooling water temperature, and cooling time, which are affected by the sprue, runner, and gate dimensions. The presence or absence of defect phenomena in products varies depending on the injection molding process conditions, which are typical of weld lines and air traps. A weld line refers to a line that occurs when two melt fronts come into contact at an angle of 135 degrees or less, which is vulnerable to a cracking risk when the weld line is long or it occurs simultaneously with an air trap. Therefore, the main criteria for selecting process conditions are a defect phenomenon that minimizes the possibility of cracks and a decrease in cycle time to ensure productivity.

### 2.3. Taguchi Design of Experiment Method

To achieve the goals of an experiment without first designing it, an enormous number of trials must be conducted, making it difficult to verify the optimal value. Even in the case of injection molding, if one factor affects the overall results, injection molding conditions vary, and it is difficult to quantitatively predict what effect each variable exerts on the results. Therefore, in order to determine the optimal value of the main factor, the application of the design of experiments (DOE) method is necessary.

The DOE is a technique to select various factors that affect the characteristics of a target product and conduct an experiment to find its optimal condition in a feasible manner [26]. Therefore, there is a need for selecting and designing a DOE method to efficiently find control factors and their levels that have a major impact on the characteristic values.

This study selected the Taguchi DOE method to achieve the goal of minimizing the cycle time, which has the advantage of being highly effective in analyzing the results and deriving the maximum amount of information from the minimum number of experiments. The method can significantly reduce the number of experiments compared to other DOE methods by devising experimental conditions using an orthogonal array table assuming no interaction and facilitating quality measurement according to characteristics through a signal-to-noise (S/N) ratio analysis. In this case, the experimental result data are evaluated using the loss function proposed by Taguchi, which is shown in Equation (1).
(1)L(y)=k(y−m)2 Here, L(y), y, m, and k indicate loss, quality characteristic value of product, achievement value of y, and the quality loss coefficient.

The Taguchi DOE secures quality by reducing the average expected loss during the life cycle of a product. When n is a representative quality characteristic value from a $y_{1,}y_{2},$ …, $y_{n}$ distribution scattered by noise, the expected loss is shown in Equation (2).
(2)$$L\left(y\right)=k\left[MSD\right]\;=k\left[\frac{1}{n}\right]\sum_{i=1}^{n}{\left(y_{i}-m\right)}^{2}\;=k\left[\sigma^{2}+{\left(\mu -m\right)}^{2}\right]$$ Here, L, *MSD*, σ, and m indicate expected loss, mean squared deviation, variance of quality characteristic value, and mean.

For efficient evaluation, the S/N ratio (signal-to-noise ratio), which represents the ratio of signal-to-noise, is utilized as a standard. The S/N ratio is obtained by converting MSD into a common logarithmic function, as shown in Equation (3).
(3)$$S/N=-10\log_{10}\left(MSD\right)$$ The S/N ratio is classified into three types depending on the properties of the target characteristic values: The larger the better characteristics, nominal the best characteristics, and smaller the better characteristics. In the first type, the higher the target characteristic value, the better the S/N ratio, while the third type refers to the case where the lower the target characteristic value, the better the S/N ratio. The second type (i.e., nominal) refers to the cases in which the target characteristics must reach a specific value, and the formulas for calculating the S/N ratio based on the characteristic values are shown in Equations (4)~(6) [27,28].
(4)$$S/N_{\mathrm{smaller}}=-10log\left[\frac{1}{n}\sum_{i=1}^{n}y_{i}^{2}\right]$$
(5)$$S/N_{\mathrm{larger}}=-10\log\left[\frac{1}{n}\sum_{i=1}^{n}\frac{1}{y_{i}^{2}}y_{i}^{2}\right]$$
(6)$$S/N_{\mathrm{nominal}}=-10\log\left[\frac{1}{n}\sum_{i=1}^{n}{\left(y_{i}-m\right)}^{2}\right]$$ Here, $y_{1,}$ n, and m indicate the measured characteristic value of $n^{th}$ experiments, the number of measured characteristic values, and the achievement value.

## 3. Analysis Model and Runner System

### 3.1. Shape of Stack Enclosure

The enclosure is shown in Figure 2, and as shown in Figure 3, each side is named A, B, C, D, E, and F, and face A is further divided into upside and downside with half the height of side A as the axis. The enclosure is 310 × 520 × 301 mm, with a flat plate reinforced with ribs for stiffness. Compared to the size of the overall shape, the bottom of sides B and D forms a side frame with a thickness of 10 mm, and a multi-point gate is required since the thickness is 10 mm in contrast to the flow length of at least 180 mm. Since the upside shape is a flat plate with reinforced ribs, whereas the downside shape is a rectangular skeleton, there is a need for controlling the flow through multi-point gates by designing runners and gates.

### 3.2. Design of Runner System

This study predicted and designed the flow system by applying an enclosure hot runner system and a multi-point gate system using pin-point and tab gates. Since the dimensions of the gate must be determined by the flow, the flow system excluding the gate dimensions is presented in Figure 4. and summarized in Table 1.

The thickness of the upper tab gate is determined to be a maximum of 3 mm since it cannot exceed 3.06 mm, which is 90% of the connected rib thickness of 3.4 mm. Therefore, the dimensions of the connected pin-point gates must be selected within 3 mm, and all pin-point gate dimensions are unified for ease of design. Moreover, since a multi-point gate is applied, it should be selected as 1.5 mm or 2 mm, considering the shape and location of the weld line. The dimension of less than 1.5 mm is not considered since it is less than 10% of the runner diameter of 12 mm.

The height of the upper tab gate was selected as 3 mm according to the rib thickness, and since the flow length of the runner was 180 mm, the depth and length were selected as 12 and 18 mm, respectively [29].

In the case of the lower tab gate, the thickness of the connected part is 10 mm, and the height of the tab gate is set as 5 mm in thickness. Since the thickness of the connection part is relatively high and the flow changes according to the height, it is selected by considering the unfilled and defect phenomena [30]. Furthermore, since the flow length of the runner is 488 mm, the depth and length according to the depth were selected as 16 and 24 mm, which is 1.5 times, respectively.

## 4. Physical Property and Comparison of Non-Reinforced/Reinforced mPPO

### 4.1. Test of Mechnical Properties

#### 4.1.1. Tensile Test and Strain-Stress Curve

The tensile test was performed for a total of 3 times according to the ISO 527/1A/50 test standard, and the test equipment was Instron’s Universal Testing Machine (UTM). The equipment and specifications are shown in Figure 5 and Table 2. The applied specimen is type 1A, as presented in Table 3. The non-reinforced and reinforced mPPO S-S curves according to the tensile strength test results are shown in Figure 6, and the physical properties according to the tensile test are presented in Table 4.

According to the result of the tensile test, the non-reinforced mPPO exhibited ductility, while the glass fiber-reinforced mPPO showed brittleness. In comparison between glass fiber-reinforced mPPO and non-reinforced mPPO, there was a significant increase in the tensile strength, indicating that the mechanical strength was superior, whereas there was a decrease in the strain rate. The results suggest that the addition of glass fibers resulted in the physical properties of brittleness and enhanced strength, whereas there was a reduction in the strain.

#### 4.1.2. Linear Thermal Expansion Test and Coefficient

The linear thermal expansion test was performed for a total of 3 times by Sinco’s thermal melting analyzer (TMA), as shown in Figure 7 and Table 5, and conducted in the temperature range of 23~170 °C. The results are shown in Figure 8 and Figure 9, and summarized in Table 6.

The non-reinforced mPPO has a slight difference between the horizontal and vertical directions, which is close to isotropy, whereas the glass fiber-reinforced mPPO indicated a difference of three or more times between the horizontal and vertical directions, indicating its anisotropy. Moreover, the degree of deformation in glass fiber-reinforced mPPO due to temperature increases was smaller than the non-reinforced mPPO. Ultimately, glass fiber-reinforced mPPO has less expansion based on temperature compared to non-reinforced mPPO, whereas there may be a deviation in shrinkage rate between the directions parallel and perpendicular to the resin flow front. Moreover, there is a need for caution with regard to the defect phenomena, such as weld line and air traps.

### 4.2. Test of Fluidity Properties

#### 4.2.1. The pvT Test and 2-Domain Tait Modified pvT Model Coefficient

The pvT test was performed by Goettfert’s Rheograph 75, as shown in Figure 10. and Table 7. The results are calculated by 2-domain Tait modified equations [31], which are shown in Equations (7)~(13).
(7)$$\upsilon \left(T,p\right)=\upsilon_{0}(T)[1-C\ln(1+\frac{p}{B\left(T\right)})]+\upsilon_{t}\left(T,p\right)$$
$$\mathrm{when},\;T>T_{t}(∵T_{t}\left(p\right)=b_{5}+b_{6}p$$
(8)$$\upsilon_{0}=b_{1m}+b_{2m}(T-b_{5})$$
(9)$$B\left(T\right)=b_{3m}\exp[-b_{4m}\left(T-b_{5}\right)]$$
(10)$$\upsilon_{t}\left(T,p\right)=0$$
$$\mathrm{when},\;T<T_{t}$$
(11)$$\upsilon_{0}=b_{1s}+b_{2s}(T-b_{5})$$
(12)$$B\left(T\right)=b_{3s}\exp[-b_{4s}\left(T-b_{5}\right)]$$
(13)$$\upsilon_{t}(T,p)=b_{7}\exp[b_{8}\left(T-b_{5}\right)-b_{9}p]$$


Here, $\upsilon_{0}$ refers to the specific volume at zero pressure, b_1m_, b_2m_, b_3m_, b_4m_, b_1s_, b_2s_, b_3s_, and b_4m_ are the pressure and temperature dependent variables and sensitivity of the molten/solid state resins, b_5_, b_6_ are the coefficients representing the change in $T_{t}$ according to pressure, b_7_, b_8_, b_9_ are the coefficients describing the transition patterns, and an intrinsic constant, respectively. Table 8 and Table 9. summarizes the results of calculating the model coefficients by calculating the test data with the formula, and Figure 11. exhibits the graph by applying the coefficients.

The results of the pvT test shown on the graph, revealed that the degree of increase in specific volume due to temperature rise was smaller in the glass fiber-reinforced mPPO than in the non-reinforced. This result suggests that the addition of glass fibers reduced the specific volume change according to temperature as the thermal stability increased.

#### 4.2.2. Viscosity Test and Cross-WLF Viscosity Model Coefficients

The viscosity test equipment was the same as the pvT test equipment. The test was conducted in 260, 270, 280, and 290 °C. The results are calculated by the cross-WLF viscosity model [32,33], which are shown in Equations (14)~(17).
(14)$$\eta =\frac{\eta_{0}}{1+{\left(\frac{\eta_{0}\dot{\gamma }}{\tau^{*}}\right)}^{1-n}}$$
(15)$$\eta_{0}=D_{1}\exp[-\frac{A_{1}(T-T^{*})}{A_{2}+(T-T^{*})}]$$
(16)$$A_{2}=A_{3}+D_{3}\;\times \;P$$
(17)$$T^{*}=D_{2}+D_{3}\;\times \;P$$

Here, η, η_0_, $\dot{\gamma }$, and τ^*^ refer to the melt viscosity, a constant that the viscosity approaches at a very slow shear rate, the shear rate, and the critical stress level at which the transition to shear thinning occurs, a value determined by curve fitting. Additionally, n refers to a power at high shear rates, determined by curve fitting. T, T*, P, and D_1_, D_2_, D_3_, A_1_, A_2_, A_3_ refer to the temperature based on the absolute unit K, the glass transition temperature (determined by curve fitting), the pressure, and the data fitting coefficient, respectively.

The results of calculating the model coefficients with the test data formula are shown in Table 10 and Table 11, and Figure 12 shows the graph of applying the coefficients.

According to the results shown in the graph representing the viscosity test results, the viscosity of the non-reinforced mPPO was lower than the glass fiber-reinforced mPPO in all temperature ranges, which suggests that the non-reinforced mPPO had relatively excellent fluidity during injection molding.

#### 4.2.3. Resin Melt Temperature Test

The melt temperature was performed by TA’s differential scanning calorimetry (DSC), as shown in Figure 13 and Table 12. The results are shown in Figure 14, and summarized in Table 13.

The melting temperatures of non-reinforced mPPO and the glass fiber-reinforced mPPO were determined to be 263.12 and 259.64 °C, respectively, indicating a similarity with a difference of less than 5 °C.

#### 4.2.4. Thermal Conductivity Test for Specific Heat Data

The thermal conductivity test was performed for a total of 3 times by the laser flash analyzer (LFA), as shown in Figure 15 and Table 14. The specimen for the test is shown in Table 15. and results are shown in Figure 16.

According to the graph representing the specific heat results through the thermal conductivity test, both the non-reinforced mPPO and the glass fiber-reinforced mPPO exhibited large fluctuations in the measured values at 200 °C, which may correspond to a point at which the state change starts to occur around 200 °C. This result suggests that it is appropriate to set the ejection temperature at 200 °C.

### 4.3. Comparison between Non-Reinforced and Glass Fiber-Reinforced mPPO Properties

Adding glass fibers to a resin improves mechanical strength [24]. However, depending on the interfacial bonding [34], it is necessary to compare the physical properties of the two materials to confirm the properties suitable for injection molding the stack enclosure. At this time, since the comparison is required for both mechanical strength and flow characteristics, tensile strength, linear thermal expansion coefficient, pvT model coefficient, viscosity, and melting temperature were measured and compared. The comparison results of non-reinforced mPPO and glass fiber-reinforced mPPO, according to the physical property test, are summarized as follows.

(1)The non-reinforced mPPO exhibited ductility, while the glass fiber-reinforced mPPO showed brittleness. In comparison between glass fiber-reinforced mPPO and non-reinforced mPPO, there was a significant increase in the tensile strength, indicating that the mechanical strength was superior, whereas there was a decrease in the strain rate. The results suggest that the addition of glass fibers resulted in the physical properties of brittleness and enhanced strength, whereas there was a reduction in the strain.(2)The non-reinforced mPPO exhibited ductility, while the glass fiber-reinforced mPPO showed brittleness. In comparison between glass fiber-reinforced mPPO and non-reinforced mPPO, there was a significant increase in the tensile strength, indicating that the mechanical strength was superior, whereas there was a decrease in the strain rate.(3)The degree of increase in specific volume due to temperature rise was smaller in the glass fiber-reinforced mPPO than in the non-reinforced mPPO. The addition of glass fibers resulted in the physical properties of brittleness and enhanced strength, whereas there was a reduction in the strain.(4)The viscosity of the non-reinforced mPPO was lower than the glass fiber-reinforced mPPO in all temperature ranges, which suggests that the non-reinforced mPPO had relatively excellent fluidity during injection molding.(5)The melting temperatures of non-reinforced mPPO and the glass fiber-reinforced mPPO were determined to be 263.12 and 259.64 °C, respectively, indicating a similarity with a difference of less than 5 °C.(6)Both the non-reinforced mPPO and the glass fiber-reinforced mPPO exhibited large fluctuations in the measured values at 200 °C, which may correspond to a point at which the state change starts to occur around 200 °C. This result suggests that it is appropriate to set the ejection temperature at 200 °C.

Therefore, non-reinforced mPPO exhibits excellent fluidity and mold-ability during injection molding due to its relatively low viscosity, whereas reinforced mPPO showed excellent dimensional stability since it has excellent mechanical strength (>3 fold greater) and low linear thermal expansion coefficient. Since mechanical strength is a key physical property in a stack enclosure as existing metal materials need to be replaced, it is judged as appropriate to produce it with glass fiber-reinforced mPPO as shown in Table 16, which would be suitable for manufacturing stack enclosures.

## 5. Selection of Suitable Gate according to Flow Change

The enclosure shown in Figure 2 has a flat plate thickness of 4~10 mm and a rib thickness of 3.3~4.5 mm compared to the overall shape size of A, requiring flow control through a multi-point gate. Furthermore, the side E located at the top has a flat plate shape with reinforced ribs, whereas the side F located at the bottom has a rectangular frame shape, each of which requires the application of runners and gates, respectively, and this application results in the increased length of the flow path. Therefore, runner losses should be reduced through the application of a hot-runner system.

As a result, the pin-point gate was applied as a hot-runner system. Pin-point gates are suitable as multi-point gates since they have a small residual stress near the gate and there are few restrictions on application locations. After the maximum diameter is limited to 3 mm considering the plate thickness of the enclosure, the dimension is selected between 1.5 and 3 mm.

A tab gate was applied to the edge of the enclosure for balanced filling. Since the upper tab gate is connected to a rib with a thickness of 3.4 mm, a height of 3 mm was selected, and a tab gate with a height of 5 mm was randomly modeled at the edge with a thickness of 10 mm at the bottom. However, the tab gate at the bottom has a large change in flow rate depending on the height, which is closely related to the flow of filling from the bottom. Since the time to freeze varies depending on the flow rate, the dimension must be selected considering the cycle time.

Injection molding analysis was performed using Autodesk Moldflow, a general-purpose software. Figure 17 shows the application of the flow system after meshing the enclosure with 2,943,284 tetrahedral elements. As the cooling channel does not overlap with the runner system and the product has a shape close to a right angle, a simple shape was used, and its diameter was 12 mm.

### 5.1. Injection Molding Analysis Results according to Pin-point Gate Diameter

To select the pin-point gate dimension, this study explained the injection molding analysis results based on the weld line generated due to the flow through the multi-point gate. The weld line analysis results according to four cases of pin-point gate diameters of 1.5 and 2 mm are shown in Figure 18.

According to the injection molding analysis results, the lowest angular range of 0~33.75° appeared most frequently in Figure 18a with the pin-point gate diameter of 1.5 mm. As the diameter of the pin-point gate increased as shown Figure 18a–d, the bond angle of the flow lines increased, and in Figure 18d with the pin-point gate diameter of 3 mm, the weld line range was the lowest in the range of 0~33.75° of the bond angle. Overall, the results show the distribution of weld lines with increased bond angles.

According to the results shown in Figure 18a–d, the distribution positions of the weld lines were nearly identical. In particular, the enlarged weld line, marked with a rectangle, extended to the entire C side of the molded product, which was located in a relatively thin area with a thickness of 5 mm, and the bond angle of the flow lines ranged from 0 to 33.75°.

### 5.2. Selection of Pin-Point Gate Diameter

The weld line is predicted to not occur when two or more flow fronts are in contact with each other at an angle of 135 degrees or more, and as the angle becomes narrower, the weld line seemed to appear more clearly. If the line appears clearly, it is necessary to consider whether it is in an improper position by appearance and whether there are any risks of cracking.

When the pin-point gate diameter was 1.5 mm, there was a large distribution of weld lines with the bond angles of flow lines ranging from 0 to 33.75°, which is inappropriate since it can cause multiple cracks. As the diameter of the pin-point gate increased, the distribution of bond angles with larger flow lines also increased, indicating that the narrowest distribution of weld lines occurs when the diameter is 3 mm. Moreover, in the case of the weld line extending to the entire rear surface of the molded product, since the bond angle of the flow lines was determined to be the largest when the pin-point gate diameter was 3 mm, the suitable pin-point gate diameter is 3 mm.

However, when the pin-point diameter is 3 mm, the weld line appearing on the C side is distributed on a thin plane, which is vulnerable to cracking and inappropriate in appearance. Therefore, there is a need for shifting the distribution of the weld line to the bottom edge, which is relatively thick at 10 mm and inconspicuous, by changing the flow.

To move the location of the weld line downward, the point where the flow line filled from above and the one filled from below should be moved. To control the flow, as shown in Figure 19a, the shape of the tapered lower pin-point gate from the existing gate entrance to the part leading to the runner was modeled as shown in Figure 19b by adding an untampered gate length. The results are presented in Figure 20.

According to the analysis results, as shown in Figure 20, the position of the weld line that had previously extended to the entire C side was moved downward, and it was disconnected at the midpoint, thereby reducing the possibility of crack occurrence. Therefore, the diameter of the pin-point gate is selected as 3 mm, and the shape of the pin-point gate at the bottom is selected as shown in Figure 19b, to which the untampered section was partially applied.

### 5.3. Selection of Tab Gate

The height of the tab gate connected to the 3.4 mm thick rib at the top was selected as 3 mm. However, in the case of the tab gate connected to the 10 mm thick rib at the bottom, there is a significant change in flow rates with height, and the flow filled from the bottom varies with flow rate. Therefore, to determine whether the height of the tab gate affects the above identified aspect of the weld line at the C side, this study conducted an analysis on the 5 and 6.5 mm tab gates before and after applying the pin-point gate diameter of 3 mm and modifying the shape, respectively, and the results are presented in Figure 21.

According to the analysis results, when the height of the tab gate is 6.5 mm, as shown in Figure 21a,b, the range of the contact angle of the flow front becomes relatively wider, while the shape of the weld line on the C side is continuously long in both (a) and (b).

When the height of the tab gate is 6.5 mm, as shown in Figure 21c,d, the range of the flow front becomes relatively wider, and in both (c) and (d), the weld line appears in a shape where the center of the C plane is broken, while the shape of the weld line on the C side disconnected at the midpoint in both (c) and (d).

Therefore, when the height of the tab gate changed from 5 to 6.5 mm, the contact angle range of the weld line widened without any shape change.

Moreover, to determine the influence of the cycle time by the height of the tab gate, the result on the time to reach ejection temperature is presented in Figure 22.

The time to reach ejection temperature was 142.9 and 144.4 s, respectively, as shown in Figure 22a,c, where the tab gate height was 5 mm. The time was 129.4 and 132.6 s, respectively, as shown in Figure 22b,d.

Therefore, when the shape of the pin-point gate was the same but the height of the tab gate was shifted from 5 to 6.5 mm, the time to reach ejection temperature decreased by 12.89 and 11.80 s, respectively.

When simultaneously considering the results of the weld line, the height of the tab gate had a greater effect on the cycle time than the change in the aspect of the weld line. Therefore, this study presents an analysis by selecting the cycle time reduction as a control factor of the DOE method.

As a result, this study selected the diameter and shape of the pin-point gate based on the analysis results, and in the case of the tab gate, it selected the tab gate connected to the 3.4 mm thick rib at the top as 3 mm. Furthermore, the analysis results revealed that the thickness of the tab gate connected to the 10 mm thick rib at the bottom exerted a greater effect on the cycle time than the weld line. Therefore, this study intends to conduct an analysis by selecting the thickness of the tab gate as a control factor of the DOE method. The selection results are summarized in Table 17.

## 6. Analysis of Injection Molding by DOE

### 6.1. Setting DOE Factors and Their Levels

To minimize the cooling time, this study designates the time to reach ejection temperature as the characteristic value; the smaller the better. Since the cooling time refers to the time to reach ejection temperature of resin in the cavity, it is related to the mold temperature and the melting temperature of the resin. This study selected the height of the tab gate as a control factor. Since the mold temperature is typically set to a value within the range of 70~100 °C, 80 °C was selected in this range. The resin melting temperature is typically set to a value in the range of 10 °C or higher from the melting temperature point of the material. Therefore, 270, 280, and 290 °C were selected as values obtained by adding 10 to 260 °C based on the measurements of properties. Furthermore, the previous analysis results confirmed that the height of the tab gate affected the time to reach ejection temperature. Therefore, the tab gate height was chosen as a control factor here, with 7 mm divided into three levels, as thickness is recommended to increase by 1 mm from 5 mm to 75% of the thickness. Table 18 and Table 19 summarize the control factors and their levels and the experimental plan by setting factors, respectively.

The conditions of the DOE method, excluding the control factors, are summarized in Table 20 with a filling time of 6 s, V/P switchover of 98%, packing pressure of 80% of injection pressure for 1 s, and coolant temperature of 20 °C.

### 6.2. Injection Molding Analysis Result

The filling results are presented in Figure 23 to confirm the reliability of the injection analysis results depending on each level of combination.

According to the analysis, the filling time was found to be 4.672~5.905 s, and in Figure 23, it was (a) 5.139 s, (b) 5.442 s, (c) 5.642 s, (d) 5.005 s, (e) 5.227 s, (f) 5.515 s, (g) 4.481 s, (h) 4.672 s, (i) 4.989 s, indicating no occurrence of unfilled portions. This suggests that the analysis results according to the DOE method are reliable.

In addition, Figure 24 presents the analysis results of the weld line due to the multi-point gate.

According to the results of the weld line analysis in accordance with the level combination, a 60~100° weld line appeared at the bottom of the A side, as shown in Figure 24a, and a 0~60° weld line appeared at the top right of the A side. A 15~60° weld line was distributed across the surface at the bottom of the C side. Overall, multiple weld lines ranging from 60 to 101° were distributed.

In the case of Figure 24b, a 70~120° weld line appeared at the bottom of the A side, and a 0~100° weld line appeared at the top right of the A side. A 10~55° weld line was distributed on both sides at the bottom of the C side, where the disconnection of weld lines was observed at the midpoint. Overall, multiple weld lines of 70~110° were distributed.

In the case of Figure 24c, a 75~100° weld line appeared at the bottom of the A side, and a 30~100° weld line appeared at the top right of the A side. A 30~55° weld line was distributed on both sides at the bottom of the C side, where the disconnection of weld lines was observed at the midpoint. Overall, multiple weld lines of 65~120° were distributed.

In the case of Figure 24d, a 60~100° weld line appeared at the bottom of the A side, and a 20~100° weld line appeared at the top right of the A side. A 10~50° weld line was distributed on both sides at the bottom of the C side, where the disconnection of weld lines was observed at the midpoint. Overall, multiple weld lines of 65~110° were distributed.

In the case of Figure 24e, a 70~110° weld line appeared at the bottom of the A side, and a 30~100° weld line appeared at the top right of the A side. A 10~50° weld line was distributed on both sides at the bottom of the C side, where the disconnection of weld lines was observed at the midpoint. Overall, multiple weld lines of 60~120° were distributed, which is relatively small.

In the case of Figure 24f, a 60~100° weld line appeared at the bottom of the A side, and a weld line appeared at the top both sides of the A side; however, it is close to 135°. A 10~40° weld line was distributed on both sides at the bottom of the C side, where the disconnection of weld lines was observed at the midpoint. Overall, multiple weld lines of 50~120° were distributed.

In the case of Figure 24g, a 70~100° weld line appeared at the bottom of the A side, and a 60~80° weld line appeared at the top right of the A side. Overall, multiple weld lines of 65~120° were distributed.

In the case of Figure 24h, a 40~80° weld appeared at the top right of the A side. Overall, multiple weld lines of 60~120° were distributed, which is relatively small.

In the case of Figure 24i, a 80~110° weld line appeared at the bottom of the A side, and a 40~90° weld line appeared at the top right of the A side. A 60~125° weld line was distributed on both sides at the bottom of the C side, where the disconnection of weld lines was observed at the midpoint. Overall, multiple weld lines of 60~130° were distributed.

As shown by the above results, the weld line is greatly influenced by the shape and size of the gate rather than being biased toward one process variable. As shown in Figure 24, the disconnection was observed at the midpoint of the weld lines, whereas there was nearly no disconnection in Figure 24g,h. This result suggests that the flow has been improved by modifying the shape of the pin-point gate, and the optimal value of control factors by S/N ratio calculation can be selected since there is a low possibility of a weld line appearing across the C side at the level value of each control factor.

To confirm that the level of each factor to minimize the production process was reasonable in performing the experimental design method, the injection time and weld line results were confirmed. In nine cases, when checking the result of the injection time, it was found to be from 4.672 to 5.905 s; in particular, short molding did not occur in all cases. In addition, from the weld line results in all nine cases, the weld line on the C side was moved or broken to a safe site, and the possibility of crack generation decreased as the angle of the weld line increased. Therefore, considering that short molding did not occur and the position of the weld line was safe, it was judged as appropriate to perform the experimental design method.

The target characteristic value of the DOE method is the time to reach ejection temperature, which can be determined from the analysis results shown in Figure 25 and Table 21.

The time to reach ejection temperature was 118.4~305.0 s, with a minimum time of 118.4 s, which occurred in Case 4 with the melt temperature of 280 °C, mold temperature of 80 °C, and tab gate’s height of 6 mm as the condition of the control factor.

### 6.3. Results of Injection Molding Analysis and Proposal of Process Conditions

A characteristic value is better as the time to reach ejection temperature becomes shorter. Therefore, Table 22 summarizes the calculated results using Equation (2.4), where the smaller the better characteristics are applied. The results are summarized in Figure 26 and Table 23.

The effects on time to reach ejection temperature showed significance in order of melt temperature, mold temperature, and tab gate’s height, as shown in Table 23. The optimal level is the maximum value of S/N ratio, respectively, which was melt temperature of 270 °C, mold temperature of 80 °C, and tab gate’s height of 6 mm.

### 6.4. Verification of Proposed Injection Molding Process Conditions

Verification analysis was conducted by equally applying the finally selected factor results, as well as other process conditions, and no unfilled sections were observed. To check the filling flow of the main weld line points at the C side, where cracks can occur, as previously confirmed, Figure 27 displays the filling flow of the C side in a chronological order.

According to the analysis results, filling and packing of the enclosure were completed in 5.527 s, and a constant flow was filled without stagnant sections during filling.

As shown in Figure 27e, according to the filling result at the C side, since the two flow lines are in contact with each other at the bottom, the weld line can be predicted to occur at the edge with a thickness of 10 mm or more at the bottom.

Figure 28a displays the filling time including packing, and Figure 28b displays the time to reach ejection temperature; therefore, it represents the total cycle time by adding both results.

The filling time including packing was 5.527 s, and the time to reach ejection temperature was 102.1 s (i.e., 16.3 s shorter than the lowest time of 118.4 s of the DOE method) corresponding to the cooling time. The cycle time per enclosure production is predicted to be 107.627 s by adding the filling time of 5.527 s and the cooling time of 102.1 s.

Figure 29a,b summarizes the weld line and air trap results. As shown in Figure 29a, the distribution of 30~35° weld line was mainly observed, while there was a decrease in the distribution of 0~30° weld line. When comparing the top right and lower edges of the A side, where weld lines are widely distributed, with the air trap shown in Figure 29b, the air trap was found to be 0.5 or less, and the joint edge of the C and E sides where the air trap was strongly generated exhibited a distribution of 70~135°, indicating that the possibility of cracking is low.

When the melt temperature of 270 °C, mold temperature of 80 °C, and tab gate’s height of 6 mm selected through the DOE method were applied, no unfilled portions occurred, the time to reach ejection temperature was 102.1 s, and the cycle time was predicted to be 107.627 s, showing a decrease in weld line distribution. Therefore, this study proposes the final flow system prediction and injection molding process conditions, as shown in Table 24 and Table 25.

### 6.5. Strength Analysis Result

To verify the strength of the stack enclosure, to which the mPPO developed in this study was applied, strength analysis was conducted through Ansys-Static, a general-purpose software tool. To proceed with the strength analysis, the enclosure was divided into 446,312 elements, and the lower part was fixed as a constraint condition. In addition, the pressure was directly applied to the rigid plate area of 402 × 399 mm in a cross-section at the top of the stack enclosure. In this manner, the force was applied to the point where 146 MPa, the ultimate strength of glass fiber-reinforced mPPO, was generated. The result of the reaction force appearing in the enclosure is presented in Figure 30 and the deformation result is presented in Figure 31.

According to the analysis results, when a pressure of 0.40 MPa was applied, the maximum stress of 145.48 MPa occurred at the point shown in Figure 30, and a deformation of 26.368 mm occurred in the (−)*Z*-axis direction in Figure 31. Therefore, the maximum mass that the stack enclosure can withstand was determined to be 5933 kg based on the maximum stress. Since the weight of one sport utility vehicle is 1800 kg, it is possible to mount both the fuel cell and related hardware in the stack enclosure.

### 6.6. Discussion of Results

#### 6.6.1. Prototype of Stack Enclosure

To verify the analysis results, the injection mold process was performed according to the conditions in Table 25 and the runner system in Table 24. The mold is shown in Figure 32, and the prototype product is shown in Figure 33.

As a result of injection molding, the total cycle time, including mold opening/closing time and resin melting time, was 190 s, and it can be confirmed that molding was performed well without cracks on the outer surface. In addition, its weight was 3.65 kg.

#### 6.6.2. Discussion of Overall Results

(1)Compared non-reinforced and glass fiber-reinforced mPPO, which developed in previous research, non-reinforced mPPO exhibits excellent fluidity and mold-ability during injection molding due to its relatively low viscosity, whereas reinforced mPPO showed excellent dimensional stability since it has excellent mechanical strength (>3-fold greater) and low linear thermal expansion coefficient [20,23,24]. Since mechanical strength is a key physical property in a stack enclosure as existing metal materials need to be replaced, it is judged as appropriate to produce it with glass fiber-reinforced mPPO, which would be suitable for manufacturing stack enclosures.(2)A runner system was predicted for injection molding of glass fiber-reinforced mPPO. As a multi-point gate application, pin-point gate and tab gate were proposed with hot-runner. When a multi-point gate is applied, since the number of flow lines increases and weld lines inevitably occur, it is important to move the weld line to a location where cracks are less likely to occur by controlling the flow. In the case of the enclosure, a weld line crossing the surface was formed on the back side (face C), and the pin-point gate diameter was selected to reduce the weld line in consideration of the risk of cracks. Therefore, the location of occurrence has been moved.(3)The Taguchi DOE was used to select the optimal conditions to improve productivity compared to the existing methods. The cooling time that occupies more than 70% of the cycle time, namely, the time to reach ejection temperature, was set as the target characteristic value. At this time, mold temperature and resin melt temperature, which are closely related to the time for the temperature of the melt to solidify, were selected as control factors. The tab gate’s height was selected as a control factor, which affects the flux related to solidifying the resin.(4)As a result of the analysis according to the level combination, the level with the maximum S/N ratio for each control factor was melt temperature of 270 °C, mold temperature of 80 °C, and tab gate’s height of 6 mm.(5)Verification analysis was conducted by equally applying the finally selected factor results. As a result, no unfilled portions occurred, the time to reach ejection temperature was 102.1 s, and the cycle time was predicted to be 107.627 s, showing a decrease in weld line distribution. It is expected that weight and material costs can be reduced using mPPO compared to die-casting manufacturing using existing aluminum, and production costs can be reduced by securing productivity through reducing cycle time. The comparison results are shown in Table 26. In addition, as a result of the strength analysis, it can withstand the load 5933 kg.(6)Therefore, the selected conditions are judged to be appropriate, and the finally proposed flow system and injection molding process conditions are shown in Table 27 and Table 28.

## 7. Conclusions

In this study, the following procedure was carried out: First, development of mPPO and presentation through physical property tests; second, prediction of the injection molding process flow system for stack enclosure production; third, proposal of injection molding process conditions to secure productivity; fourth, verification of conditions through mechanical stiffness analysis, and fifth, presentation of the conclusions.

In summary, the properties of non-reinforced/reinforced mPPO (PPO 40%/PA66 60%) developed in previous studies were compared. Although the fluidity of the reinforced mPPO was relatively poor, the mechanical strength was three times better. Thereafter, simulations were conducted using actual physical properties to propose the flow system and process conditions. The analysis results suggested the shape and dimensions of the pin-point gate and the tab gate considering the weld line. Moreover, injection molding process conditions that minimized the processing time were proposed through the Taguchi design of the experiment. The simulations were performed to confirm the mechanical strength of the stack enclosure to which the corresponding physical properties were applied. They presented that no damage occurred up to 5933 kg. Finally, it was verified through the prototype, proving that the proposed runner system and process conditions were suitable.

Further research is needed to verify the developed mPPO through various tests, such as acceleration, vibration, and environmental tests, in order to apply to mass production.

## Figures and Tables

**Figure 1 polymers-15-01303-f001:**
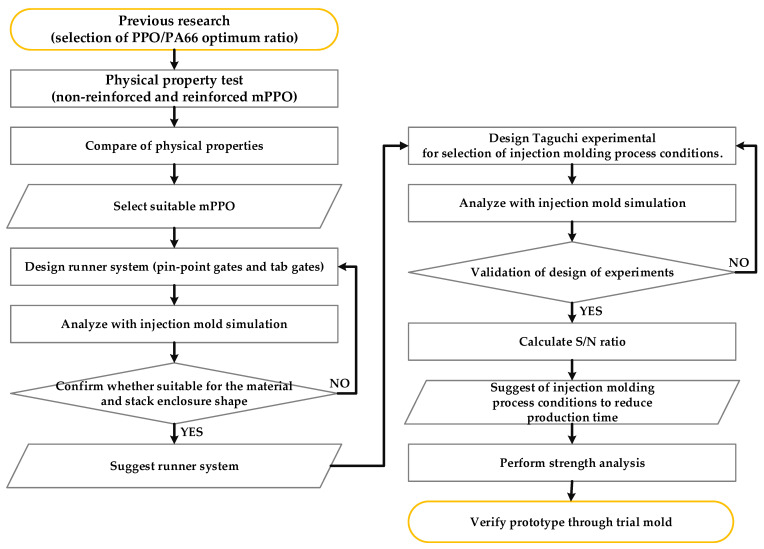
Flow chart of the study.

**Figure 2 polymers-15-01303-f002:**
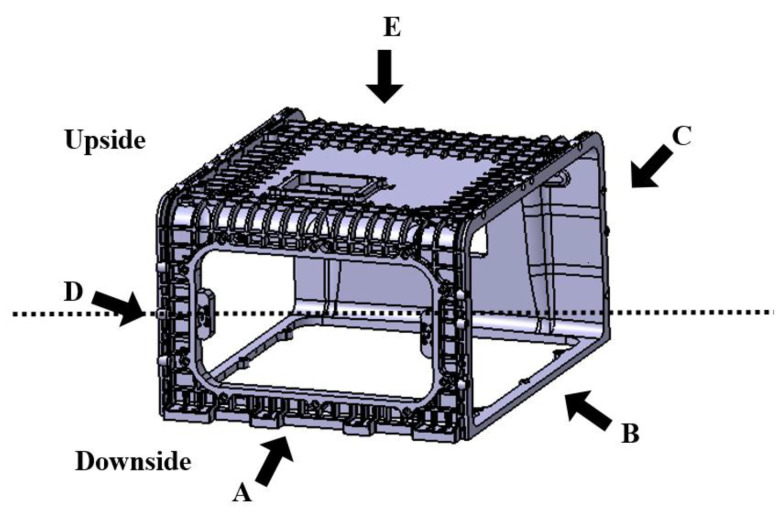
Overall shape of stack enclosure indicated for each visual point of view with A, B, C, D, E, F, upside and downside.

**Figure 3 polymers-15-01303-f003:**
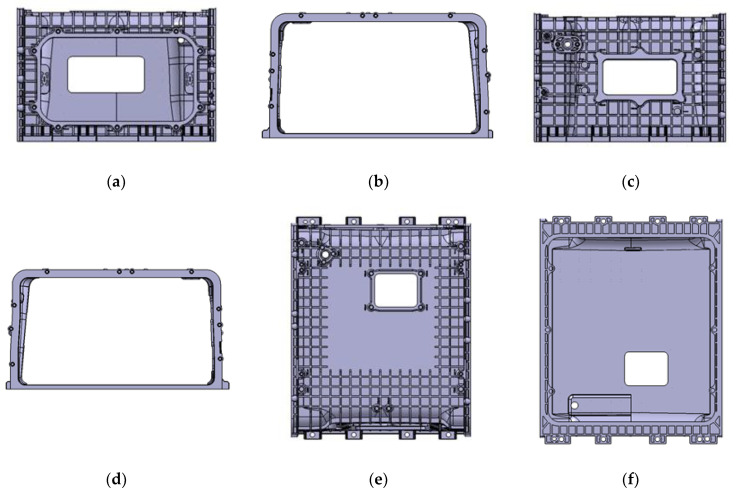
Title of stack enclosure’s faces. (**a**) Face A: Front view; (**b**) Face B: Side view 1; (**c**) Face C: Back view; (**d**) Face D: Side view; (**e**) Face E: Upside view; and (**f**) Face F: Bottom view.

**Figure 4 polymers-15-01303-f004:**
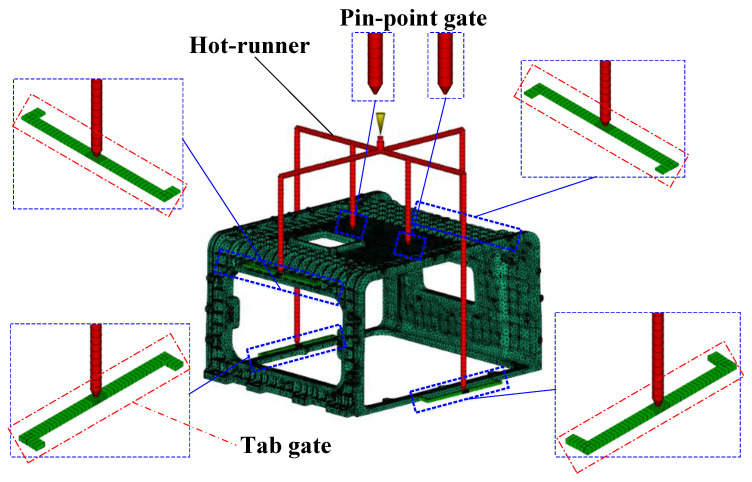
Stack enclosure runner system design. Hot-runner, pin-point gates, and tab gates.

**Figure 5 polymers-15-01303-f005:**
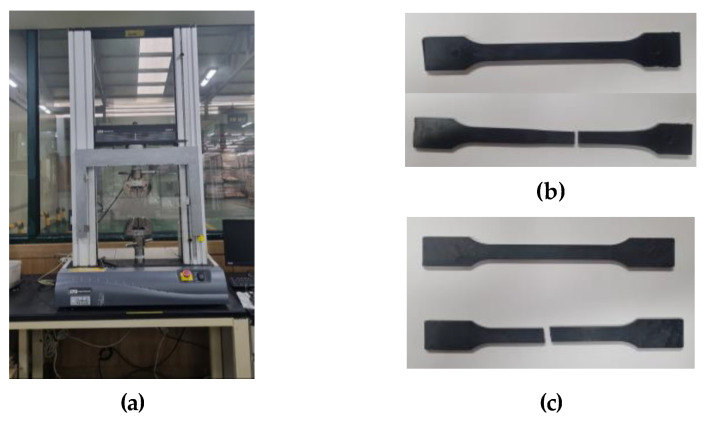
Tensile test. (**a**) Universal testing machine, specimen test result of (**b**) non-reinforced mPPO and (**c**) reinforced mPPO.

**Figure 6 polymers-15-01303-f006:**
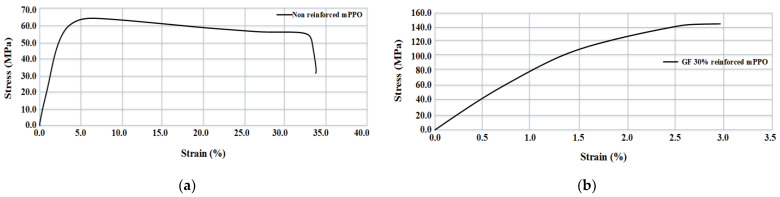
Strain-stress curve. (**a**) Non-reinforced mPPO; (**b**) GF-reinforced mPPO.

**Figure 7 polymers-15-01303-f007:**
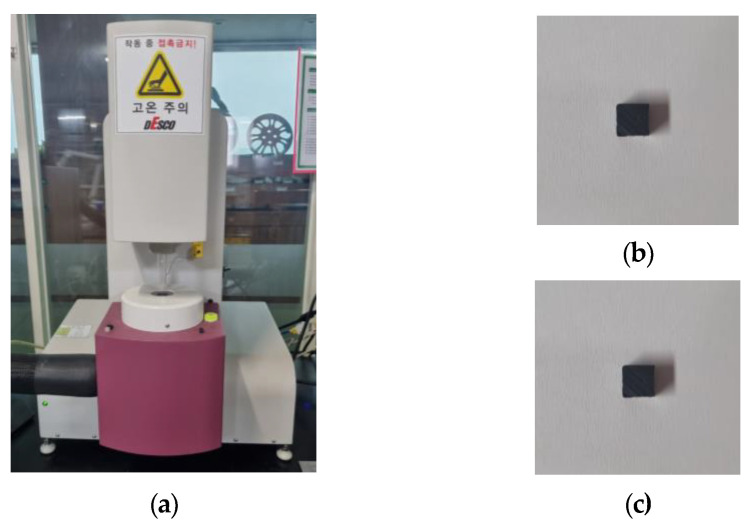
Thermal expansion test. (**a**) Thermal melting analyzer, specimen test result of (**b**) non-reinforced mPPO, and (**c**) reinforced mPPO.

**Figure 8 polymers-15-01303-f008:**
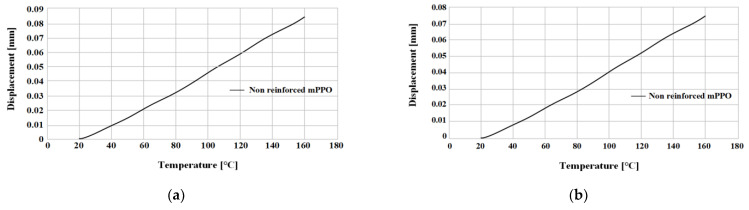
Displacement by thermal expansion from 20 to 165 °C test. Non-reinforced mPPO: (**a**) Horizontal; (**b**) vertical.

**Figure 9 polymers-15-01303-f009:**
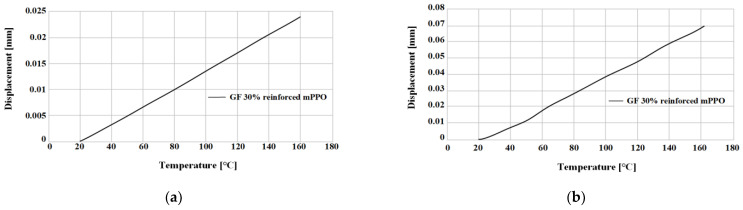
Displacement by thermal expansion from 20 to 165 °C test. GF-reinforced mPPO: (**a**) Horizontal; (**b**) vertical.

**Figure 10 polymers-15-01303-f010:**
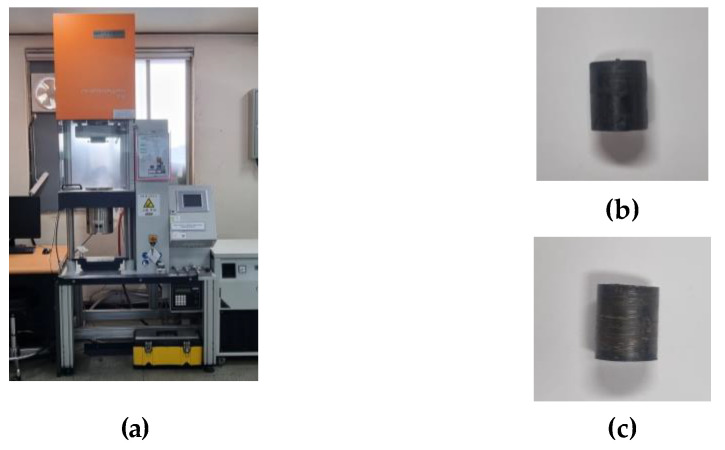
The pvT test. (**a**) Rheograph for pvT and viscosity test, specimen test result of (**b**) non-reinforced mPPO and (**c**) reinforced mPPO.

**Figure 11 polymers-15-01303-f011:**
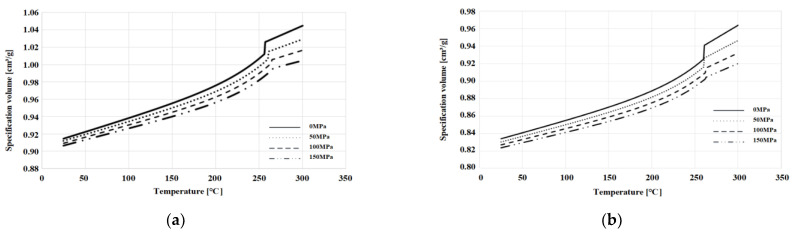
The 2-domain modified Tait pvT model: 0, 50, 100, and 150 MPa for (**a**) non-reinforced mPPO; (**b**) GF-reinforced mPPO.

**Figure 12 polymers-15-01303-f012:**
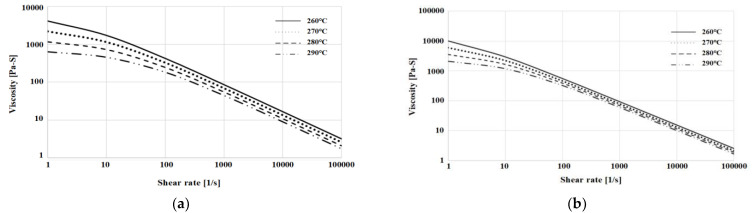
Cross-WLF viscosity: 260, 270, 280, and 290 °C for (**a**) non-reinforced mPPO; (**b**) GF-reinforced mPPO.

**Figure 13 polymers-15-01303-f013:**
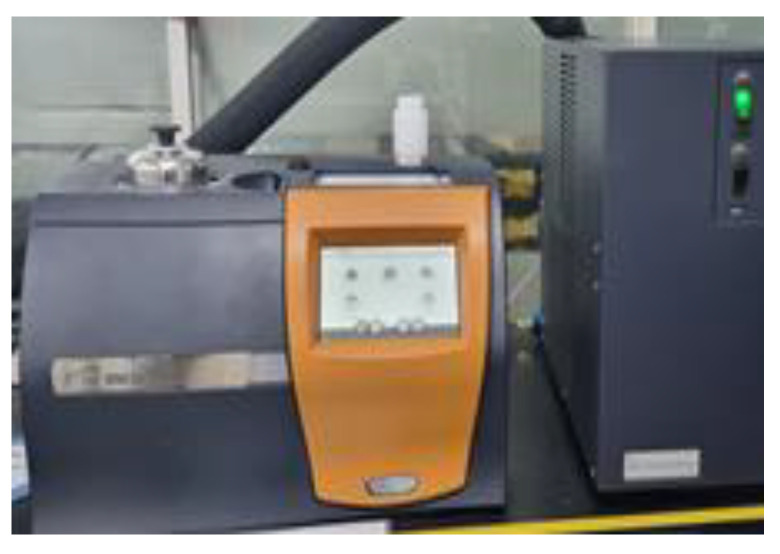
Differential scanning calorimetry.

**Figure 14 polymers-15-01303-f014:**
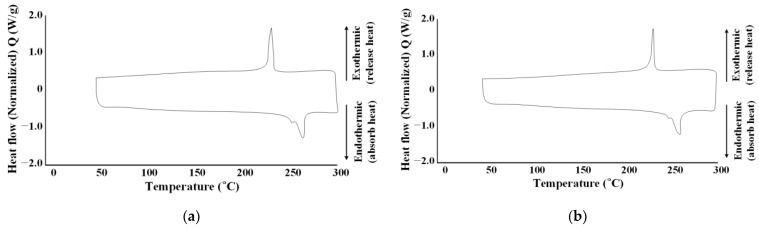
Melt temperature graph. (**a**) Non-reinforced mPPO; (**b**) GF-reinforced mPPO.

**Figure 15 polymers-15-01303-f015:**
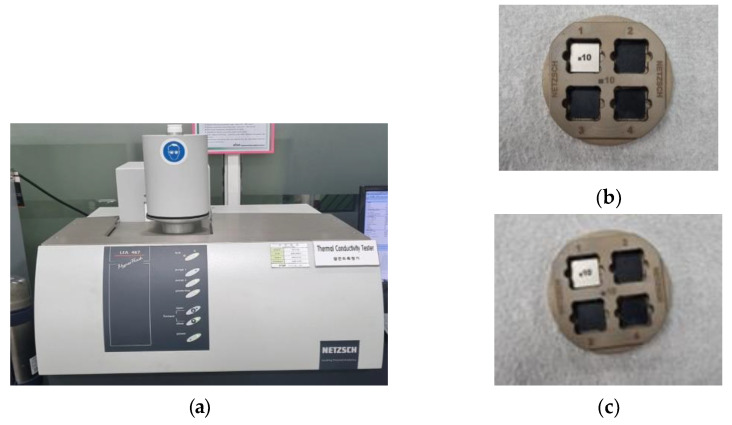
Thermal conductivity test. (**a**) Laser flash analyzer, specimen test result of (**b**) non-reinforced mPPO and (**c**) reinforced mPPO.

**Figure 16 polymers-15-01303-f016:**
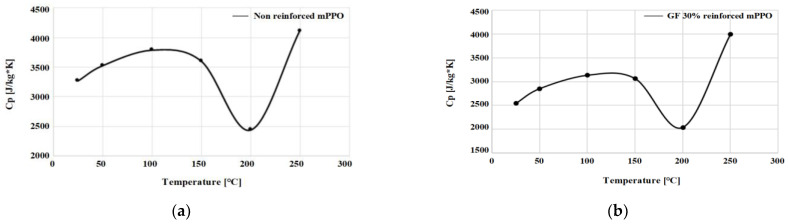
Specific heat in 25, 50, 100, 150, 200, and 250 °C: (**a**) Non-reinforced mPPO; (**b**) GF-reinforced mPPO.

**Figure 17 polymers-15-01303-f017:**
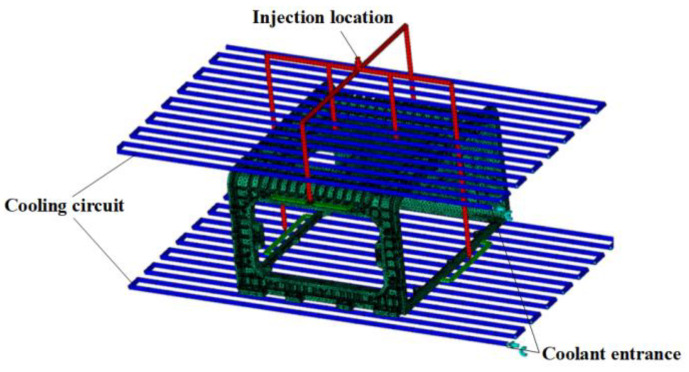
Meshing.

**Figure 18 polymers-15-01303-f018:**
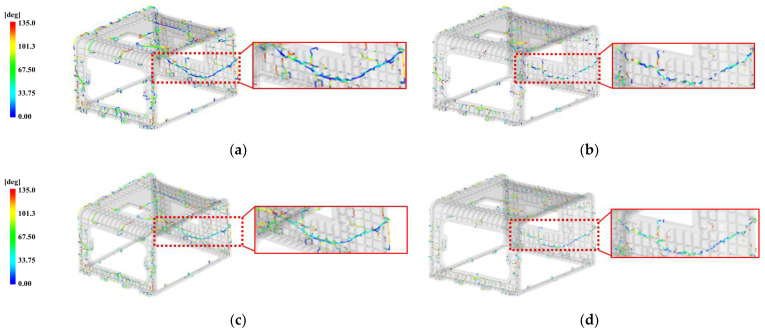
Weld line by the pin-point gate size on face C. (**a**) 1.5 mm; (**b**) 2 mm; (**c**) 2.5 mm; (**d**) 3 mm.

**Figure 19 polymers-15-01303-f019:**
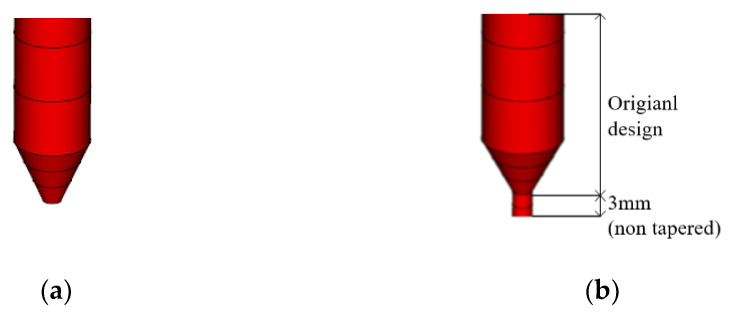
Pin-point gate shape. (**a**) Original design; (**b**) non-tapered modified design.

**Figure 20 polymers-15-01303-f020:**
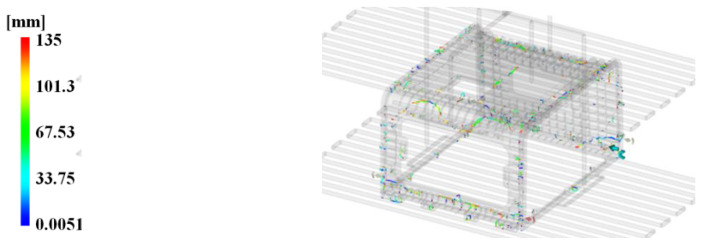
Overall weld line result with modified pin-point gate shape.

**Figure 21 polymers-15-01303-f021:**
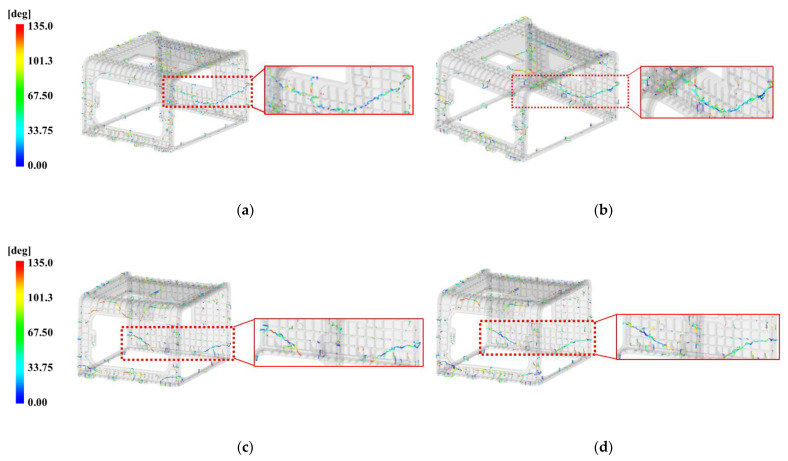
Weld line effects on face C with tab gate’s height. (**a**) Original pin-point gate shape and tab gate’s height 5 mm; (**b**) original pin-point gate shape and tab gate’s height 6.5 mm; (**c**) modified pin-point gate shape and tab gate’s height 5 mm; (**d**) modified pin-point gate shape and tab gate’s height 6.5 mm.

**Figure 22 polymers-15-01303-f022:**
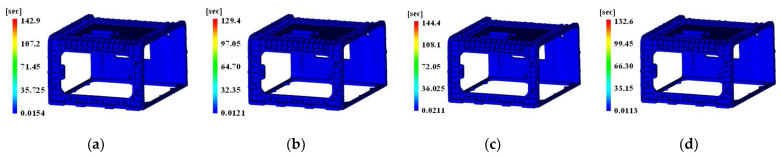
Time to reach ejection temperature effects with tab gate’s height. (**a**) Original pin-point gate shape and tab gate’s height 5 mm; (**b**) original pin-point gate shape and tab gate’s height 6.5 mm; (**c**) modified pin-point gate shape and tab gate’s height 5 mm; (**d**) modified pin-point gate shape and tab gate’s height 6.5 mm.

**Figure 23 polymers-15-01303-f023:**
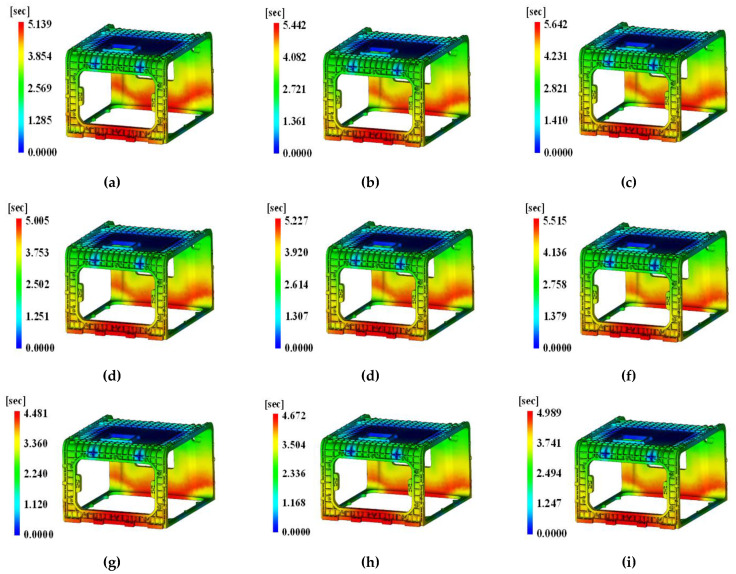
Filling time results of overall view with DOE simulation number. (**a**) Case 1; (**b**) Case 2; (**c**) Case 3; (**d**) Case 4; (**e**) Case 5; (**f**) Case 6; (**g**) Case 7; (**h**) Case 8; (**i**) Case 9.

**Figure 24 polymers-15-01303-f024:**
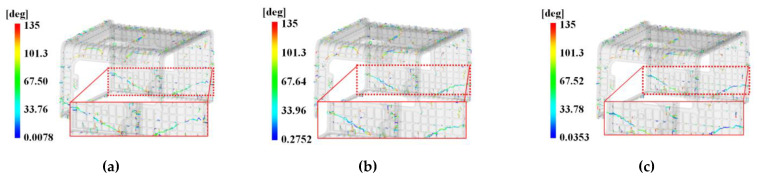
Weld line results specifically analyzed for face C with DOE. (**a**) Case 1; (**b**) Case 2; (**c**) Case 3; (**d**) Case 4; (**e**) Case 5; (**f**) Case 6; (**g**) Case 7; (**h**) Case 8; (**i**) Case 9.

**Figure 25 polymers-15-01303-f025:**
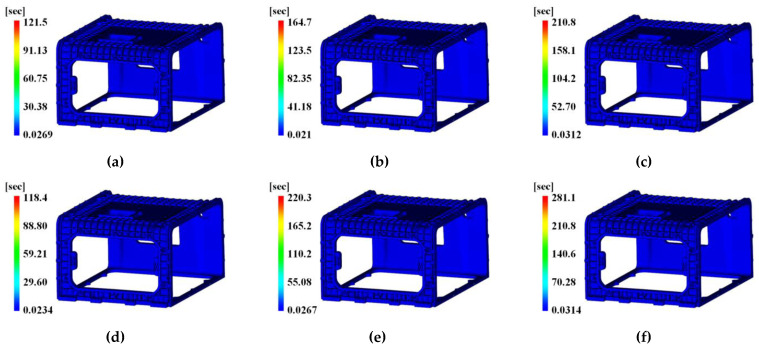
Time to reach ejection temperature results with DOE simulation number. (**a**) Case 1; (**b**) Case 2; (**c**) Case 3; (**d**) Case 4; (**e**) Case 5; (**f**) Case 6; (**g**) Case 7; (**h**) Case 8; (**i**) Case 9; the earliest time is represented on the surface.

**Figure 26 polymers-15-01303-f026:**
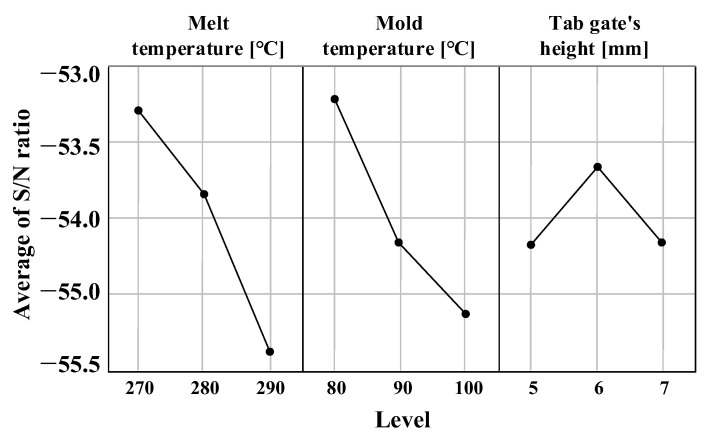
Average S/N ratio of control factor represented by graph.

**Figure 27 polymers-15-01303-f027:**
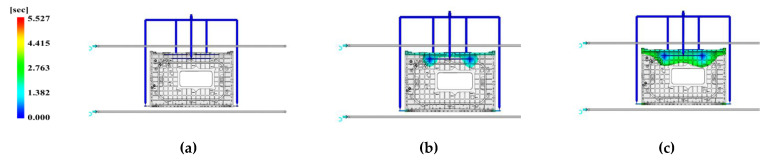
Filling time represented on the viewpoint of Face C. Time of (**a**) 0.2303 s; (**b**) 1.151 s; (**c**) 2.994 s; (**d**) 3.6884 s; (**e**) 5.066 s; (**f**) 5.527 s.

**Figure 28 polymers-15-01303-f028:**
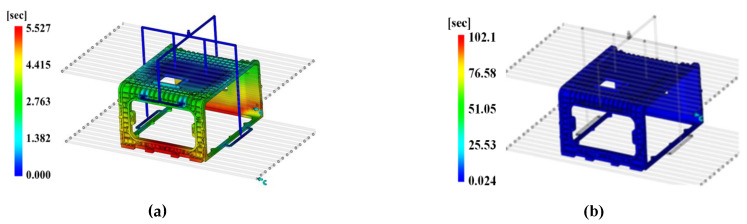
Total cycle time. (**a**) Injection time; (**b**) time to reach ejection temperature.

**Figure 29 polymers-15-01303-f029:**
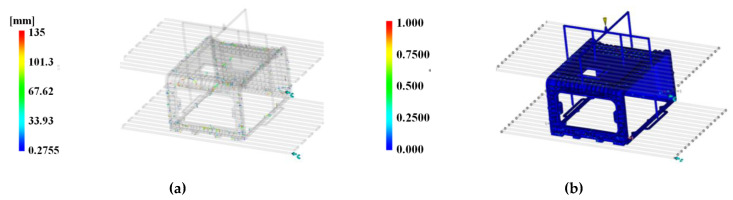
Defect results. (**a**) Weld line; (**b**) air trap.

**Figure 30 polymers-15-01303-f030:**
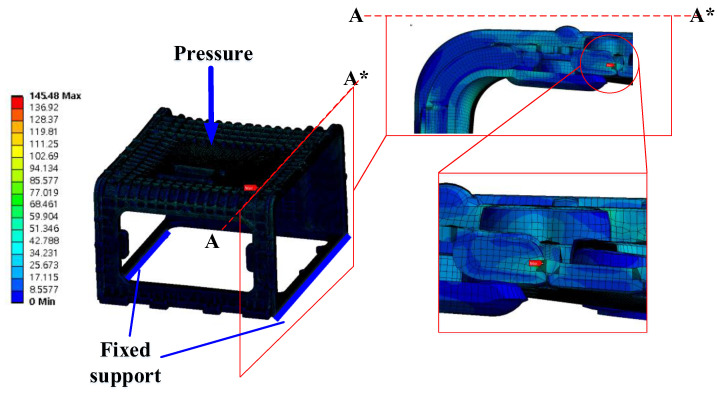
Equivalent stress results. Maximum point.

**Figure 31 polymers-15-01303-f031:**
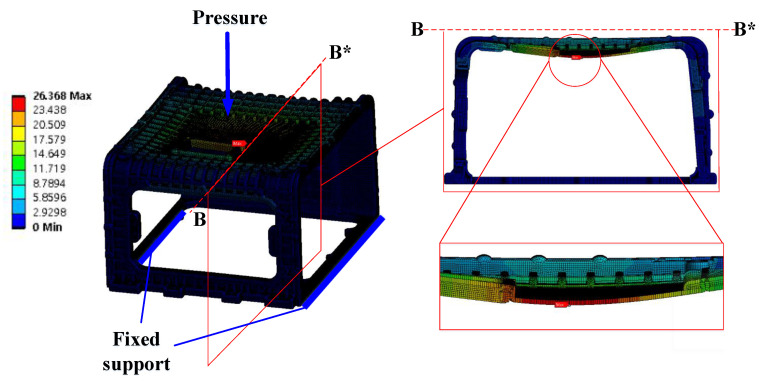
Deformation results. Maximum point in side view.

**Figure 32 polymers-15-01303-f032:**
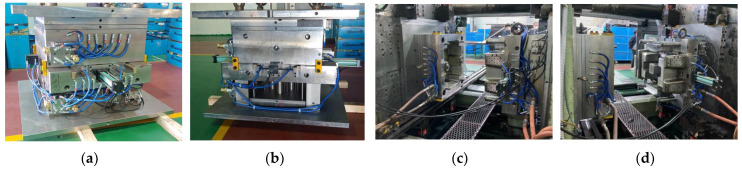
Mold of the stack enclosure. (**a**) Closed mold by side 1; (**b**) closed mold by side 2; (**c**) mounted on the injection machine; (**d**) mounted on the injection machine on the other side view.

**Figure 33 polymers-15-01303-f033:**
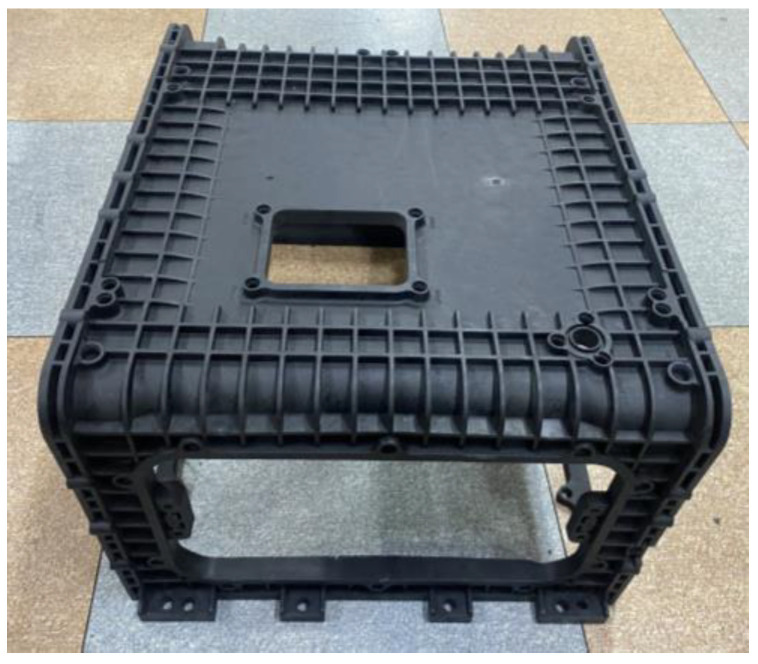
The prototype of the stack enclosure.

**Table 1 polymers-15-01303-t001:** Runner system size. Pin-point gate’s diameter, tab gate size, and runner size.

Description		Value
Pin-Point Gate’s Diameter [mm]		Undetermined (1.5, 2, 2.5, 3)
Tab gate [mm], upside	Height	3
	Width	12 (runner size)
	Length	18 (width × 1.5)
Tab gate [mm], downside	Height	5
	Width	16 (runner size)
	Length	24 (width × 1.5)
Runner [mm]	Length	180 ~ 488
	Diameter	12 (upside), 16 (downside)

**Table 2 polymers-15-01303-t002:** Universal testing machine specification.

Description	Value
Force capacity	10 kN
Crosshead travel	1172 mm
Vertical test space	1242 mm
Horizontal test space	420 mm
Maximum speed	508 mm/min
Minimum speed	0.05 mm/min
Maximum return speed	610 mm/min
Footprint dimensions (h×w×d)	1610 × 760 × 710 mm
Position control resolution	9.9 mm
Frame axial stiffness	38 kN/mm
Maximum force at full speed	10 kN
Maximum speed at full force	508 mm/min
Weight	122 kg
Maximum power requirement	730 VA

**Table 3 polymers-15-01303-t003:** Tensile strength test specimen specification.

Specimen Shape	Description	Value
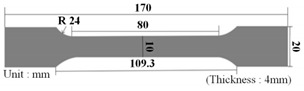	Length	80 mm
	Width	10 mm
	Thickness	4 mm
	Notch angle	45°
	Notch radius	0.25 mm

**Table 4 polymers-15-01303-t004:** Tensile strength test results.

Description	mPPO (Non-Reinforced)	mPPO (GF-Reinforced)
Poisson’s rate	0.37	0.38
Young’s modulus	2650 MPa ± 21	8845 MPa ± 32
Yield strength (0.2% offset)	5.344 MPa ± 0.76	17.883 MPa ± 0.97
Ultimate strength	64 MPa ± 3	146.7 MPa ± 6

**Table 5 polymers-15-01303-t005:** Thermal melting analyzer specification.

Description	Value
Thermal range	−90~800 °C
Temperature programmer	0.1~60 °C/min
Isothermal Stability	±0.4 °C
Maximum sample size	up to 10 mm (in length)
Furnace Winding	Ni-chrome
Furnace movement	by electric motor

**Table 6 polymers-15-01303-t006:** Coefficient of thermal expansion from 20 to 165 °C test.

mPPO	Coefficient (1 µmm/m °C)	
	Horizontal	Vertical
Non-reinforced	121.369 ± 3.019	103.951 ± 3.112
GF-reinforced	31.717 ± 1.88	97.183 ± 1.76

**Table 7 polymers-15-01303-t007:** Rheograph specification.

Description	Value
Temperature	up to 400 °C
Speed	up to 40 mm/s
Press	up to 1000 bar
Drop height	0.61 m
Impact velocity	3.46 m/s
Dimensions (width*depth*height)	660 × 380 × 840 mm
Weight	110 kg

**Table 8 polymers-15-01303-t008:** The mPPO (non-reinforced) 2-domain modified Tait pvT model coefficients.

Coefficients	Unit	Value	Coefficients	Unit	Value
b5	K	536.15	b1s	m^3^/kg	0.0009874
b6	K/Pa	$2.50\;\times \;10^{-7}$	b2s	m^3^/kg·K	$3.0741\;\times \;10^{-7}$
b1m	m^3^/kg	0.001025	b3s	Pa	$1.1213\;\times \;10^{8}$
b2m	m^3^/kg·K	$4.374\;\times \;10^{-7}$	b4s	1/K	0.014524
b3m	Pa	$6.6156\;\times \;10^{12}$	b7	m^3^/kg	$4.898\;\times \;10^{-5}$
b4m	1/K	0.00	b8	1/Pa	0.04248
			b9	1/Pa	$8.4575\;\times \;10^{-9}$

**Table 9 polymers-15-01303-t009:** The mPPO (GF-reinforced) 2-domain modified Tait pvT model coefficients.

Coefficients	Unit	Value	Coefficients	Unit	Value
b5	K	518.15	b1s	m^3^/kg	0.001570
b6	K/Pa	$2.49\;\times \;10^{-7}$	b2s	m^3^/kg·K	$2.694\;\times \;10^{-7}$
b1m	m^3^/kg	0.001125	b3s	Pa	$5.589\;\times \;10^{8}$
b2m	m^3^/kg·K	$5.6725\;\times \;10^{-7}$	b4s	1/K	0.01680
b3m	Pa	$1.107\;\times \;10^{9}$	b7	m^3^/kg	$3.8103\;\times \;10^{-5}$
b4m	1/K	0.001136	b8	1/Pa	0.04124
			b9	1/Pa	$8.8223\;\times \;10^{-9}$

**Table 10 polymers-15-01303-t010:** The mPPO (non-reinforced) Cross-WLF viscosity model coefficients.

Coefficients	Unit	Value
n	-	0.2816
τ*	Pa	$1.6518\;\times \;10^{+4}$
D1	Pa-s	$3.65\;\times \;10^{+18}$
D2	K	413.15
D3	K/Pa	0
A1	-	48.65
A2	K	51.6

**Table 11 polymers-15-01303-t011:** The mPPO (GF-reinforced) Cross-WLF viscosity model coefficients.

Coefficients	Unit	Value
n	-	0.2168
τ*	Pa	$2.1853\;\times \;10^{+4}$
D1	Pa-s	$1.72\;\times \;10^{+19}$
D2	K	408.15
D3	K/Pa	0
A1	-	48.65
A2	K	51.6

**Table 12 polymers-15-01303-t012:** Differential scanning calorimetry specification.

Description	Value
Temperature range	Room temperature ~725 °C
Temperature accuracy	±0.1 °C
Temperature precision	±0.05 °C
Flow accuracy	1.0%

**Table 13 polymers-15-01303-t013:** Melt temperature results.

Description	mPPO (Non-Reinforced)	mPPO (GF-Reinforced)
Melt temperature [°C]	263.12 ± 2.57	259.64 ± 1.02

**Table 14 polymers-15-01303-t014:** Thermal conductivity analyzer specification.

Description	Value
Temperature range	−100~500 °C
Maximum output	10 J/pulse
Thermal diffusivity measurement range	0.01~1000 mm^2^/s
Thermal conductivity measurement range	0.01~2000 W/mK

**Table 15 polymers-15-01303-t015:** Specimen for thermal conductivity test.

Transverse	Length	Thickness
10 mm	10 mm	1 mm

**Table 16 polymers-15-01303-t016:** Specimen for thermal conductivity test.

Description	Mechanical Property		pvT				Viscosity		Tm
Value	Poisson’s rate	0.38	b5	518.15 K	b1s	0.001570 m^3^/kg	n	0.2168	260 °C
	Young’s modulus	8845 MPa	b6	$2.49\;\times \;10^{-7}$ K/Pa	b2s	$2.694\;\times \;10^{-7}$ m^3^/kg·K	τ*	$2.185\;\times \;10^{+4}$ Pa	
	Yield strength	17.88 MPa	b1m	0.001125 m^3^/kg	b3s	$5.589\;\times \;10^{+8}$ Pa	D1	$1.72\;\times \;10^{+19}$ Pa-s	
	Ultimate strength	146.7 MPa	b2m	$5.673\;\times \;10^{-7}$ m^3^/kg·K	b4s	0.01680 1/K	D2	408.15 K	
	CTE	0.38 µmm/m °C	b3m	$1.107\;\times \;10^{+9}$ Pa	b7	$3.810\;\times \;10^{-5}$ m^3^/kg	D3	0 K/Pa	
			b4m	0.001136 1/K	b8	0.04124 1/Pa	A1	48.65	
					b9	$8.822\;\times \;10^{-9}$ 1/Pa	A2	51.6 K	

**Table 17 polymers-15-01303-t017:** Runner system size determined by injection mold analysis results.

Description		Value
Pin-point gate’s diameter [mm]	Upside	3
	Downside	3 (with non-tapered shape)
Tab gate [mm], upside	Height	3
	Width	12 (runner size)
	Length	18 (width×1.5)
Tab gate [mm], downside	Height	5 (undetermined)
	Width	16 (runner size)
	Length	24 (width×1.5)
Runner [mm]	length	180~488
	diameter	12 (upside), 16 (downside)

**Table 18 polymers-15-01303-t018:** Design of control factor and level for Taguchi experimental method.

Factor	Description	Level		
		1	2	3
A	Melt temperature [°C]	270	280	290
B	Mold temperature [°C]	80	90	100
C	Tab gate’s height [mm], downside	5	6	7

**Table 19 polymers-15-01303-t019:** L_9_(3^3^) orthogonal array with melt temperature (A), mold temperature (B), and tab gate’s height (C).

Simulation No.	A	B	C
1	1	1	1
2	1	2	2
3	1	3	3
4	2	1	2
5	2	2	3
6	2	3	1
7	3	1	3
8	3	2	1
9	3	3	2
	(1)		

**Table 20 polymers-15-01303-t020:** Conditions of injection molding except for control factors.

Description	Value
Injection time	6 s
V/P switchover	98%
Packing pressure	80% of injection pressure for 1 s
Coolant temperature	20 °C

**Table 21 polymers-15-01303-t021:** Injection molding analysis results of L_9_(3^3^) orthogonal array.

Case No.	Time to Reach Ejection Temperature [s]
1	121.5
2	164.7
3	210.8
4	118.4
5	220.3
6	281.1
7	248.3
8	299.9
9	305.0

**Table 22 polymers-15-01303-t022:** S/N ratio results of L_9_(3^3^) orthogonal array.

Case No.	Time to Reach Ejection Temperature [S]	Time to Reach Ejection Temperature S/N Ratio
1	121.5	−52.4960
2	164.7	−53.3435
3	210.8	−54.1650
4	** *118.4* **	−***52.431*** ^1^
5	220.3	−54.3251
6	281.1	−55.2850
7	248.3	−54.7804
8	299.9	−55.5616
9	305.0	−55.6351

^1^ The maximum value of S/N ratio.

**Table 23 polymers-15-01303-t023:** S/N ratio results of each control factor.

Description		Melt Temperature	Mold Temperature	Tab Gate’s Height
Level	1	− ** *53.33* **	− ** *53.24* **	−54.45
	2	−54.01	−54.41	− ** *53.80* **
	3	−55.33	−55.03	−54.42
Delta		1.99	1.79	0.64
Rank		1	2	3

**Table 24 polymers-15-01303-t024:** Design of runner system size determined by DOE results.

Description		Value
Pin-point gate’s diameter [mm]	Upside	3
	Downside	3 (with non-tapered shape)
Tab gate [mm], upside	Height	3
	Width	12 (runner size)
	Length	18 (width × 1.5)
Tab gate [mm], downside	Height	6
	Width	16 (runner size)
	Length	24 (width × 1.5)
Runner [mm]	Length	180~488
	Diameter	12 (upside), 16 (downside)

**Table 25 polymers-15-01303-t025:** Final conditions of injection molding by DOE results and calculation of S/N ratio.

Description	Value
Melt temperature	270 °C
Mold temperature	80 °C
Injection time	6 s
V/P switchover	98%
Packing pressure	80% of injection pressure for 1 s
Coolant temperature	20 °C

**Table 26 polymers-15-01303-t026:** Comparison with existing stack enclosure.

Description	Aluminum (AC4CH)	mPPO
Weight	9.5 kg	3.65 kg
Cycle time	30 min/1 ea	107.63 s/1 ea

**Table 27 polymers-15-01303-t027:** Suggestion design of runner system.

Description		Value
Pin-point gate‘s diameter [mm]	Upside	3
	Downside	3 (with non-tapered shape)
Tab gate [mm], upside	Height	3
	Width	12 (runner size)
	Length	18 (width×1.5)
Tab gate [mm], downside	Height	6
	Width	16 (runner size)
	Length	24 (width ×1.5)
Runner [mm]	Length	180~488
	Diameter	12 (upside), 16 (downside)

**Table 28 polymers-15-01303-t028:** Suggestion conditions of injection molding.

Description	Value
Melt temperature	270 °C
Mold temperature	80 °C
Injection time	6 s
V/P switchover	98%
Packing pressure	80% of injection pressure for 1 s
Coolant temperature	20 °C

## Data Availability

Not applicable.

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
