# Peer review of "A Study on mPPO Development and Injection Molding Process for Lightweight Stack Enclosure of FCEV"

_polymers, 2023, doi:10.3390/polym15051303_

Round 1

Reviewer 1 Report

Lee et al. proposed a study about developing a lightweight stack enclosure using injection-molded MPPO. I think the topic is of paramount importance following the principles of carbon neutrality. Despite this, I think this article can be published in Polymers after major revision. Some comments should be addressed as follows:

1.      The title reports two times the words “fuel cells” because FV in the acronym is the same.

2.      All the references need to be checked carefully; they are cited starting from n.19 in section 2.1.

3.      In section 2.1, quantitative previous results about PPO and MPPO materials must be reported more extensively because the state of the art is poor. Furthermore, also literature regarding injection molding of PPO material is not exhaustive. Instead, an extensive explanation of Taguchi method is presented; I think it is a well-known method, and a quick presentation with a reference is enough.

4.      Figures 3.1 and 3.2 and the relative captions should be modified to clarify the different sides of the stack.

5.      Also, Fig 3.3. is unclear; it should be enlarged.

6.      To improve the reader's understanding of the paper, the authors could incorporate a figure that illustrates the workflow of the current study.

7.      Was the mPPo prepared by the authors? In which way? Or was it supplied by some manufacturer?

8.      Were the tensile specimens manufactured by authors? How many samples were realized?

9.      Figure 4.2 and table 4.2 could be unified.

10.  What about standard deviation in materials tests? The results (curves and tables) do not have any information about this.

11.  Rather than showing equipment pictures (fig. 4.1, 4.4, and 4.7), which are traditional laboratory instruments, a picture of realized specimens, also after the tensile tests, could be more interesting.

12.  The addition of glass fibers into materials is a known aspect already presents in literature. Thus results n. 1,2,3 of section 4.2.4 are not really innovative. What are the meanings to evidence them?

13.  In the Moldflow material database, is the PPO material included? And what about mPPO? Did you use real materials for simulations?

14.  Even if the authors report that fabrication and injection tests will be performed in the future (lines 579-581), I think a paper presenting only simulation results is unreliable. Do you have any further results?

15.  Some typos are presented in the paper at different lines.

Reviewer 2 Report

The authors developed PPO and proposed injection molding process for Lightweight Stack Enclosure of FCEV Fuel Cells. Overall, the study is interesting and important. However, this article should be majorly revised prior to its publication.

1. It seems that the references were not included in the introduction part. Please check.

2. For all of the equations written, please justify all of the parameters throughout the text.

3. The conclusions are too much. It should be clear and consize. Should be no figures as well.

4. The figure captiond should be explained as clear as possible. So many figure captions but the description are not specific.

5. And also the Table captions.

6. No citations in the discussion part, thus this article lacks of scientific rigor.

7. The authos just showed their results without deep discussion and comparison to other works.

Reviewer 3 Report

In this work, the runner system with pin-point gate and tab gate’s sizes are proposed. In addition, injection molding process conditions were proposed with the results of cycle time 107.627 seconds and reduced weld lines. The results developed FCEV and mPPO with injection molding for replacing existing material.

Some detailed comments:

*It is recommended that the authors give a table of performance parameters or references for materials.

*In the introduction part- I did not understand the author's reasons for choosing mPPO (not other polymers) to replace the existing metal materials of the fuel cell stack enclosure of FCEV.

*In the introduction part- The authors did not give any relevant research progress by other researchers.

*Where are references 1-18?

* Why not choose other types of filler? Such as carbon fiber, PBO fiber, PBI fiber, M5 fiber, PI fiber.

*Why didn't the authors conduct real experiments to verify the correctness of the calculation results?

*I suggest the author increase the size and clarity of the images.

In conclusion, this article is recommended to be major revised.

Reviewer 4 Report

1. Delete 1)2)3)4) in the abstract.

2. In Introduction the first paragraph is too long, simplify the introduction of carbon neutrality.

3. For this purpose, this study will 1) develop mPPO and present it through physical property tests, 2) predict the injection molding process flow system for stack enclosure production, 3) propose injection molding process conditions to secure productivity, and 4) verify conditions through mechanical stiffness analysis.

Delete 1)2)3)4) in the sentence.

4. In 2.2 Injection Molding Process and Its Major Factors, the number of formula is discontinuous, two formula 2.3, no formula 2.5. What the meaning of characters in these formulas should be given.

5. The color of Fig. 3.3 is dark, it is difficult to see the details in the figure. Fig 6.10Fig 6.11 and Fig 6.12 is the same.

6. In table 3.1 Pin point gates diameter , seems wrong.

7. In 4.4.1 please give the full name of UTM.

8. Conclusion part should not include tables.

9. Conclusions are too long, the conclusions should be simplified. 

Round 2

Reviewer 1 Report

Thanks to the authors, they answered all my questions satisfactorily, and the quality of the paper is now improved.

Reviewer 2 Report

The authors have tried to revise the manuscript according to my comments.

The caption in Fig 5.1 seems too short (Only meshing?). Please make it more specific.

After that, I think the manuscript will be ready for publication.

Reviewer 3 Report

The article can be accepted.

Reviewer 4 Report

This paper can be accepted in present form.